# The origin of the boundary strengthening in polycrystal-inspired architected materials

Chen Liu 📖 [1]✉, Jedsada Lertthanasarn[1] & Minh-Son Pham 📖 [1]

Crystal-inspired approach is found to be highly successful in designing extraordinarily damage-tolerant architected materials. i.e. meta-crystals, necessitating in-depth fundamental studies to reveal the underlying mechanisms responsible for the strengthening in meta-crystals. Such understanding will enable greater confidence to control not only strength, but also spatial local deformation. In this study, the mechanisms underlying shear band activities were investigated and discussed to provide a solid basis for predicting and controlling the local deformation behaviour in meta-crystals. The boundary strengthening in polycrystal-like meta-crystals was found to relate to the interaction between shear bands and polygrain-like boundaries. More importantly, the boundary type and coherency were found to be influential as they govern the transmission of shear bands across meta-grains boundaries. The obtained insights in this study provide crucial knowledge in developing high strength architected materials with great capacity in controlling and programming the mechanical strength and damage path.

---

[1] Engineering Alloys, Department of Materials, Imperial College London, London, UK. ✉email: c.liu16@imperial.ac.uk

Architected materials built on ordered arrangements of internal structures are lightweight and excellent at bearing load and absorbing impact energy, enabling their wide applications in the fields of aerospace, automobiles, medical devices, packaging and infrastructures[1–4]. Lattice materials formed by a periodic arrangement of unit cells consisting of a regular network of struts are one particular type of architected materials[5,6]. Extensive and in-depth works have been carried out with a focus on the elastic, plastic yield and toughness properties of singly oriented lattice materials made of different base materials[7,8], various unit cells types, such as iso-truss[9], octet-truss[10,11], body-centred cubic[12–15], gyroid[16] and auxetic[17], and fabrication techniques that enable extraordinary properties of hierarchical lattice materials[18–21]. One of the major issues for singly oriented lattices is the undesirable severe drops in strength after plastic yield due to the formation of dominant and fast propagating shear bands throughout the whole structure, in particular for stretching-dominated lattice materials made of elastoplastic materials or containing fabrication defects[22–26]. As a consequence, the strength and energy absorption capacity of lattice materials is substantially reduced, limiting the applicability of the materials for structural applications. The formation of the shear band is inherently related to the plastic yield, bending or buckling of constituent struts[27]. The uniform periodic arrangement of struts throughout the whole structure causes a consequence that once a strut collapses, the other struts of a given orientation collapse in the same way, causing the fast propagation of damage and substantial loss in strength[6,28].

A transformative approach of designing new both high strength and extraordinarily damage-tolerant architected materials, coined meta-crystals, was recently proposed to successfully translate the metallurgical mechanisms, in particular the grain boundary hardening, found in polycrystalline materials to eliminate the post-yield collapse and significantly strengthen lattice materials by designing polygrain-like structures[29]. The introduction of polycrystalline-inspired features to break the long-range periodicity in a singly oriented lattice into multiple domains named meta-grains of different orientations inhibits the fast propagation of the shear band and prevents catastrophic failure. The study demonstrates that the strength and stability of architected materials can be dramatically increased thanks to the significantly shortening and impeding effects of meta-grain boundaries on the dominant shear band. Moreover, the previous study clearly showed that the yield strength of meta-crystals increases with a reduction in the size of meta-grains, a similarity to the well-known Hall–Petch relationship found for polycrystalline metals[30–34]. Although the strengthening effects via reducing meta-grain size was clearly presented[29], key mechanisms underlying the observed strengthening were not discussed. An in-depth understanding of such mechanisms is crucial in controlling and programming not only the strength but also the local spatial deformation of meta-crystals. To unravel the underlying mechanisms, it is necessary to study the shear-band activities and the interaction between the shear bands and the boundaries between meta-grains. The study of shear bands also helps to better understand the anisotropy of lattice materials, and to answer first two key remained questions: why were the shear band formed in the ⟨1 0 1⟩{0 0 2} system as reported in ref. [29]? Are there other systems that might be active? In addition, the study of the interactions between meta-grain boundaries and shear bands helps to answer other key questions: how is the boundary between meta-grains able to stop a shear band? Similar to the grain boundary strengthening in polycrystals[33,35,36], do the characteristics, such as coherency, of boundaries affect the stopping effect? Understanding of the underlying mechanisms and answers to the questions will provide the designers more confidence and a sound

basis to fully utilise the meta-crystal approach to develop light-weight and high-performing architected materials with controllable properties.

In this study, we studied the mechanisms underlying the shear-band formation and identified active shear-band systems in a lattice structure of face-centred cubic (FCC). The buckling analyses were done to reveal the origin of shear-band formation, to explain and predict the shearing activities in FCC meta-crystals. Subsequently, meta-crystals mimicking polycrystals containing numbers of meta-grains ranging from 1 to 64 were designed and tested to study the shear-band propagation and interactions with meta-grain boundaries. The role of meta-grain boundaries and its coherency on shearing activities, yield strength and hardening were revealed and discussed.

## Results

**Shearing behaviour in singly oriented meta-crystals.** We firstly designed and tested single meta-crystals with three different orientations (i.e. 30°, 45° and 60°) regarding the loading directions parallel to global Z axis ("Methods" and Supplementary Fig. 1). The three meta-crystals were subsequently named FCC30, FCC45 and FCC60 (Fig. 1a). Figure 1b shows corresponding stress–strain responses, where clear drops in strength after yielding occurring at strain from ~3 to 10% were observed in all three lattices. This softening is due to the collapse of struts along a specific direction, forming parallel shear bands shown in Fig. 1c–e in consistent with previous reports on singly oriented lattice materials[22–25]. The relationship between shear bands and lattice orientations were further revealed in DIC analyses. Frame insets of oriented unit cells indicate the shear direction in relation to the orientation of the unit cell of the infilled structure (Fig. 1c–e). Similar to the definition of slip systems consisting of slip plane and slip direction in crystalline alloys, shear systems in meta-crystals can be defined by the plane and direction of shear. The shear systems in the designed FCC meta-crystals were

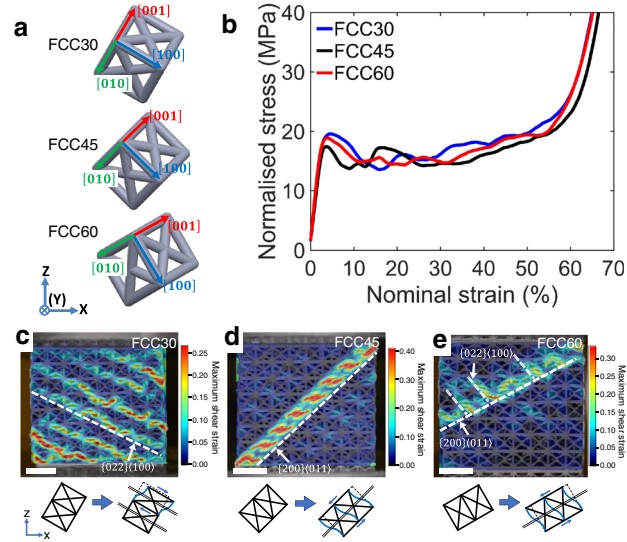

**Fig. 1 Experiment results of singly oriented FCC-inspired meta-crystals. a** The three lattice orientations of singly oriented meta-crystals: FCC30, FCC45 and FCC60, under a global orthogonal coordinate system, where the compressive loading direction is parallel to global Z axis. **b** Corresponded stress–strain curves, where stress was normalised by respective relative densities. **c–e** Localised deformation behaviour revealed by DIC analyses at a global nominal strain of 10%—cubic frame insets were used to show the lattice orientation and shearing direction. The length-scale bar in (**c–e**) is 10 mm.

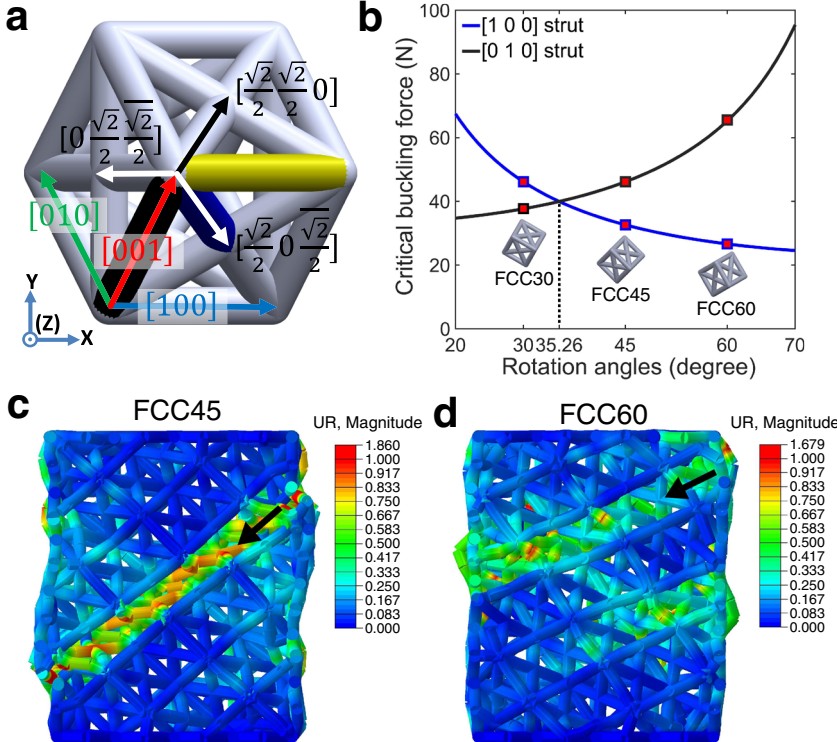

**Fig. 2 Theoretical and finite element analysis. a** Miller indices of individual struts where [1 0 0], [0 1 0] and [0 0 1] struts belong to a group of ⟨1 0 0⟩ struts, while [$\frac{\sqrt{2}}{2}$ $\frac{\sqrt{2}}{2}$ 0], [$\frac{\sqrt{2}}{2}$ 0 $\frac{\sqrt{2}}{2}$] and [0 $\frac{\sqrt{2}}{2}$ $\frac{\sqrt{2}}{2}$] struts belong to the group of ⟨$\frac{\sqrt{2}}{2}$ $\frac{\sqrt{2}}{2}$ 0⟩ struts. **b** Critical buckling force of ⟨1 0 0⟩ struts calculated for unit $K$ (effective length factor of strut) along the global $Z$ axis. **c, d** Sectional views of FEA simulations parallel to ($X$, $Y$) plane for FCC45 and FCC60 at 14.5% strain, respectively, showing rotational displacement at deformed nodes.

identified to be {0 2 2}⟨1 0 0⟩ in FCC30 and {2 0 0}⟨0 1 1⟩ in FCC45 and FCC60. It should be noted that previous studies on various singly oriented lattices reported only a single shear plane system, such as {0 2 2} plane for BCC[24], F2CCz[37] and BCCZ[38] and {0 0 2} for simple cubic[39]. This study shows that FCC meta-crystal has at least two different shearing systems. This difference highlights the importance of loading conditions to the activation of shear bands, i.e. anisotropy in shear-band activities. It is therefore essential for material designers to know the relationship between shear band activities and loading conditions for accurately controlling the damage propagation along specific directions and planes in lattice materials.

Buckling and plastic yield of the base material are often reported to be the main mechanisms responsible for the collapse of the strut, resulting in the initiation of shear band[27]. In this study, it was found that buckling was responsible for the strut collapse in FCC meta-crystals. To obtain in-depth understanding of the activation of these shearing systems in the meta-crystals, critical force and stress for buckling of individual struts (Fig. 2a) were calculated (see the section "Buckling force analyses" under Methods section) and analysed. For the unit cell used in this study, constructed struts can be classified into two groups by their orientations: ⟨1 0 0⟩ struts with slenderness ratio $S = 20$, which is larger than that of ⟨$\frac{\sqrt{2}}{2}$ $\frac{\sqrt{2}}{2}$ 0⟩ struts ($S = 14.14$). Thus, the shear behaviour is governed by the buckling of ⟨1 0 0⟩ struts, i.e. the shear plane and direction are on the planes perpendicular to ⟨1 0 0⟩ direction. Moreover, Fig. 2b shows the variation of critical buckling forces along global $Z$ axis for [0 1 0] and [1 0 0] struts with changing the orientation angle of lattice with respect to the global Y axis (see the section "Buckling force analyses"), which indicates that [0 1 0] strut buckled first in FCC30 while buckling should first occur for the [1 0 0] struts in FCC45 and FCC60.

Such buckling explains the formations of shear-band systems as experimentally observed by DIC analyses (Fig. 1c–e). In addition, a threshold orientation angle $\theta = 35.26°$ was found (Fig. 2b), where the two reported shear systems were both active. Supplementary Fig. 2 experimentally confirmed that the {0 2 2}⟨1 0 0⟩ was active and followed by {2 0 0}⟨0 1 1⟩ in the FCC single meta-crystal with $\theta = 35.26°$.

In contrast to very well-defined shear activities in FCC30 (Fig. 1c) and FCC45 (Fig. 1d), the shear bands in FCC60 appear more complex, causing difficulties in the shear-band identification. In addition to the dominant {2 0 0}⟨0 1 1⟩ shear band, some fine and localised shear bands were observed in the FCC60 lattice (Fig. 1e). This shearing behaviour was identified to be {0 2 2}⟨1 0 0⟩, which might also be another active shear-band system for FCC60. To confirm this, finite element analysis (FEA) was used to help understand the deformation behaviour of struts in this meta-crystal (Supplementary Fig. 3). FEA simulation set-up was first validated against the experimental observation for FCC45 (Fig. 2c vs Fig. 1d). The simulation shows that [1 0 0], [$\frac{\sqrt{2}}{2}$ $\frac{\sqrt{2}}{2}$ 0] and [$\frac{\sqrt{2}}{2}$ 0 $\frac{\sqrt{2}}{2}$] struts were highly rotated due to buckling, forming a localisation of deformation along {2 0 0}⟨0 1 1⟩ in agreement with the DIC observation for FCC45 (Fig. 1d), confirming the validity of simulation. The FCC60 was then simulated by using the FEA parameters identified for FCC45. Figure 2d confirms that {2 0 0}⟨0 1 1⟩ is the shear-band system in FCC60, while {0 2 2}⟨1 0 0⟩ was not seen in the FEA simulation. In addition, DIC analysis (Fig. 1e) shows that {0 2 2}⟨1 0 0⟩ was only observed locally near collapsed struts of {2 0 0}⟨0 1 1⟩. Similarly, Fig. 1c shows minute and short localisation along a different direction to the {0 2 2}⟨1 0 0⟩ in FCC30. Therefore, the shortened localisation in {0 2 2}⟨1 0 0⟩ observed in FCC60 was likely due to the local stress redistribution once a {2 0 0}⟨0 1 1⟩

shear band formed. The experimental and simulation analyses confirm that $\{0\,2\,2\}\langle1\,0\,0\rangle$ was not a shear-band system, i.e. $\{2\,0\,0\}\langle0\,1\,1\rangle$ is the only shear band system for the FCC60.

Although the origins of shear bands for designed FCC meta-crystals are clearly demonstrated from experiments and buckling analysis, it should be noted that the strength and shear behaviours of a lattice material are also influenced by the base materials, geometric dimensions (e.g. slenderness) and defects from the fabrication of unit cells[40–42]. For example, nano-sized lattice materials constructed by slender struts tend to successively buckle in layers perpendicular to the loading direction during compression, instead of forming a dominant shear band[43]. The main reason for this is due to the fact that the slender nano-sized struts were not strong to enable quick transfer and redistribution of stress, causing successive layer crumbling before the applied load could be transferred to the struts further away from compressing plates. This highlights the importance of load redistribution, in addition to the buckling, in the formation of shear bands. Further reduction in the length scale of struts can make struts stronger, hence facilitating the formation of shear bands in nano-sized lattices[40].

**Meta-grain boundary strengthening.** To reveal the effect of meta-grain boundary, we designed a meta-crystal containing two meta-grains, i.e. bi-meta-crystal, that were twinned FCC45 with the twin boundary highlighted in Supplementary Fig. 4. Figure 3a shows the stress–strain curves of bi-meta-crystal with comparison to that of a single meta-crystal. As the unit cells in each meta-grain were aligned to the same angle to the global loading direction, each meta-grain should have the same yield strength

due to buckling. Therefore, the buckling-induced yield strength of the single and bi-meta-crystals was expected to be the same. Interestingly, however, the measured yield strength and the firstly reached peak strength of the bi-meta-crystal were slightly higher than that of the single meta-crystal (Fig. 3a), indicating an influential role of the boundary between the two meta-grains. In addition, although both meta-crystals show softening behaviour after reaching the yield stress, the post-yield collapse in the bi-meta-crystal was much less severe. The hardening rate estimated by $\Delta\sigma/\Delta\varepsilon$ with $\Delta\varepsilon = 0.01$ was calculated to quantify the improvement in stabilising the deformation behaviour of meta-crystals (Fig. 3b). Positive rates represent the hardening, while negative rates represent softening. The absolute value of the largest softening rate at 34.6 MPa for the single meta-crystal was almost double than that of the bi-meta-crystal at 18.3 MPa, suggesting the stress drop was much less severe in the bi-meta-crystal. In addition, Fig. 3b shows that the rate fluctuated drastically for the single meta-crystal while the fluctuation was much smaller for the bi-meta-crystal, i.e. the meta-crystal containing two meta-grains behaves in a much more stable manner than the single meta-crystal due to the presence of the boundary between the two meta-grains. Figure 3c and d showed that a single and straight shear band in the single meta-crystal was formed along with the maximum shear stress, resulting in an abrupt drop in post-yield strength (Fig. 3a) and the hardening rate (Fig. 3b). In contrast, two diffused and shortened shear bands were formed and aligned in two different directions which were symmetrical about the twin boundary in the bi-meta-crystal (Fig. 3e, f). Shear bands in each meta-grain were identified to be of the $\{2\,0\,0\}\langle0\,1\,1\rangle$ system in consistent with the observation in the single

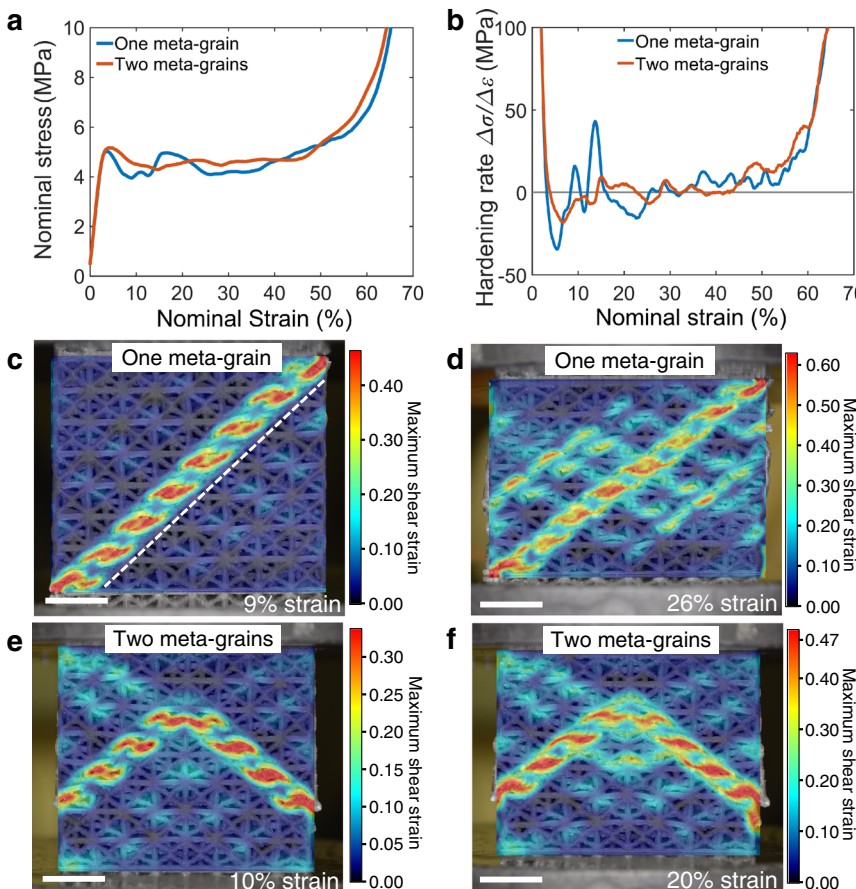

**Fig. 3 Comparison between single and bi-meta-crystals. a** Stress–strain curves. **b** Hardening rate. **c–f** Deformation behaviours of each lattice at given strains after DIC analyses. The length-scale bar in (**c–f**) is 10 mm.

meta-crystal FCC45 (presented in Fig. 1d). The two symmetrical shear bands were much shortened to almost a half of that in the single meta-crystal and confined by the twin boundary introduced between the two meta-grains, confirming the boundary was able to stop and deflect shear bands, minimising the stress drop after yielding. In addition to the shortened shear bands, the deformation in the twinned bi-meta-grains were more homogeneous and uniform with a much lower degree of localisation than in the single meta-crystals (Fig. 3f vs d)—note that the scale bar of the maximum shear strain in the bi-meta-crystal is smaller than that in the single meta-crystal. In other words, the boundary significantly stabilised the deformation after yielding in the bi-meta-crystal by inhibiting the propagation of shear bands and diffusing the localised deformation.

The strengthening effect of meta-grain boundary on meta-crystals is further highlighted by reducing the meta-grain size. It should be noted that all the meta-crystals exhibited a highly reproducible response with only minimal variations mainly seen in the starting point of densification (Supplementary Fig. 5). In addition, to minimise the effect induced by dependence on build orientation in the fabrication process[44,45], all the presented polygrain-like meta-crystals were fabricated using the same build direction which was along the global $Z$ direction ("Methods"). Figure 4a shows the normalised stress–strain curves of meta-crystals with different numbers of meta-grains. Reducing meta-grain size results in an increase in strength, which is consistent with our previous study[29]. Note that the stress–strain curves in this study include additional data points and were normalised by their relative densities to help minimising the effect induced by the relative density. In addition, it was reported that a relationship between the nominal yield strength $\sigma_{Y\_L}$ at 0.2% strain offset and the size of meta-grain that can be described by a Hall–Petch-like relationship[29], Eq. (1). A least-square linear

curve fitting of the normalised yield stresses $\sigma_{NY\_L}$ using Eq. (1) gives $\sigma_0 = 12.49\,\mathrm{MPa}$ and $k = 14.46\,\mathrm{MPa} \cdot \mathrm{mm}^{1/2}$ (Fig. 4b). The $\sigma_0 = 12.49\,\mathrm{MPa}$ is the macroscopic critical stress to initiate the yielding of a single FCC45 meta-crystal with nominally infinite size. This critical stress is smaller than the normalised yield stress (14.8 MPa) of the single FCC45 meta-crystal because the meta-crystal has a finite size. Such a linear relationship reaffirms the applicability of the Hall-Petch relationship to describe the boundary strengthening in meta-crystals.

$$\sigma_{Y\_L} = \sigma_0 + \frac{k}{\sqrt{d}} \tag{1}$$

where $d$ is the size of meta-grains, $\sigma_0$ is frictional stress and $k$ is a material constant.

The reduction in the size of meta-grains also improved post-yield strength and increasingly stabilised the flow stress of meta-crystals in agreement with Fig. 3b. Stress drops were completely eliminated in meta-crystals containing 32 and 64 meta-grains (Fig. 4a). Flow stress at a given strain (e.g. $\sigma_{\varepsilon=0.3}$ at $\varepsilon = 0.3$) was quantified and plotted against $\frac{1}{\sqrt{d}}$ to demonstrate the effect of meta-grain boundaries on the strain hardening of meta-crystals. It was found that $\sigma_0 = 8\,\mathrm{MPa}$ and $k = 43\,\mathrm{MPa} \cdot \mathrm{mm}^{1/2}$ well described the relationship between the flow stress $\sigma_{\varepsilon=0.3}$ and the size of meta-grain. $k = 43$ higher than ratio 14.46 for normalised yield strength suggests that meta-grains boundaries had more effect on the hardening in plastic deformation (Fig. 4b). Such increased flow stress as reducing the meta-grain size is related to that hardening behaviour during plastic deformation. Thus, the hardening rates during plastic deformation of meta-crystals were quantified and shown in Fig. 4c. Compared to the large fluctuations of rate change in a single meta-crystal, the hardening rates became more stable during deformation with a reduction in meta-grain size. Most notably, the stable and increased hardening

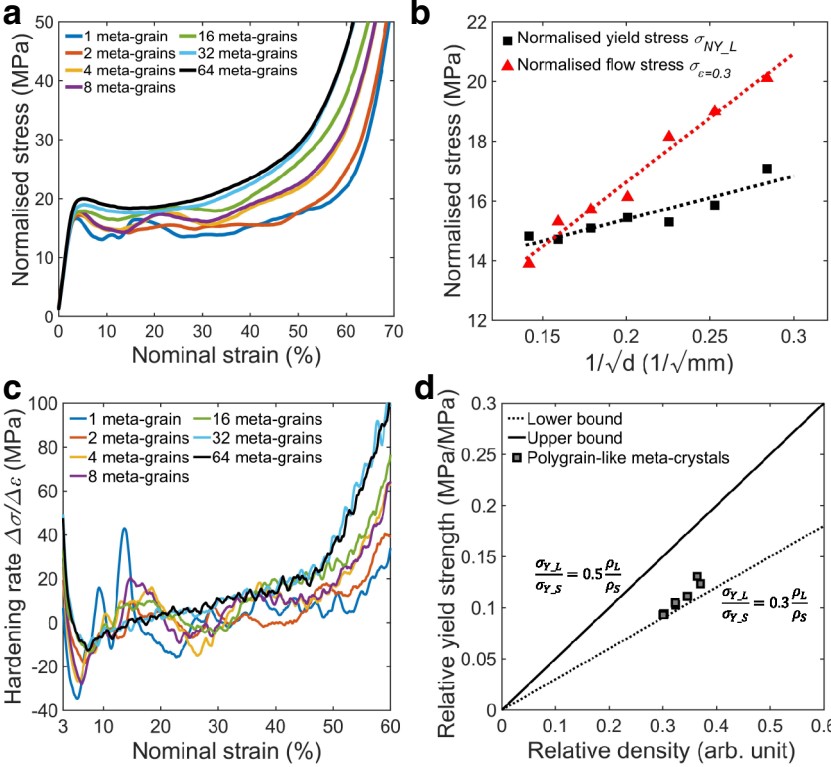

**Fig. 4 Mechanical properties of lattices with different numbers of meta-grains. a** Stress–strain curves. Note that stresses were normalised by the relative density of individual meta-crystals. **b** Relationship between normalised yield stress $\sigma_{NY\_\perp}$, flow stress $\sigma_{\varepsilon=0.3}$ at $\varepsilon = 0.3$ and meta-grain size. **c** Hardening rates. **d** Relationship between relative yield strength and relative density.

rates of all polygrain-like meta-crystals led to extended uniform plastic deformation with a continuous increase in the flow stress. Such hardening results from the interaction between meta-grain boundaries and shear bands as highlighted by DIC analyses in Fig. 3. It is worth noting that the design of stochastic architected materials also results in strain hardening, but there was a compromise in yield strength and the plastic flow stress is still serrated[28]. By contrast, the meta-crystal approach achieves both high yield strength, stable flow stress and significant hardening.

It is known that the relative density can strongly influence the mechanical properties of architected lattice materials[6]. According to Maxwell's criteria[46], the used FCC unit cell is a stretching-dominated lattice. It theoretically predicts that relative yield strength should scale linearly with relative density with the pre-factor coefficient constant varying from 0.3 to 0.5[5,21,47]. The scaling relationship between relative density and relative yield strength (the ratio of the yield strength of lattice material, $\sigma_{Y\_L}$, to that of the base material, $\sigma_{Y\_S}$) is shown in Fig. 4d. The scaling relationship for the relative strength quite follows the theoretical prediction with the pre-factor close to the lower bound of 0.3, indicating a modest dependence of the relative strength with the relative density. In addition, because the variation in the measured relative densities was relatively small (Supplementary Table 1), the normalisation of stress can significantly minimise the contributions from the relative density, hence better high-lighting the contribution from the meta-grain boundaries.

DIC analyses were used to study the propagation and spatial distribution of shear bands to reveal a more in-depth understanding of the role of meta-grain boundaries on the plastic deformation of meta-crystals. Figure 5 shows designed meta-crystal models and the shear bands distributions at 20% strain. CAD models with the frame skin removed are shown to reveal the internal lattice orientation and struts connections across meta-grain boundaries. It shows that the shear localisation in each meta-grain was dominated by lattice orientation similar to slip behaviour in crystals. A shear-band system the $\langle 0\,1\,1 \rangle$ directions on $\{2\,0\,0\}$ planes was observed in each meta-grain, which was consistent with that in the FCC45 single meta-crystal (Fig. 1d). The direction of the shear band was deflected by around 90° across the meta-grain boundary due to the change in lattice orientations across the boundary. The decrease in the size of meta-grains significantly shortened the length of shear bands

(Fig. 5a–e) in consistent with the observation of the twinned bi-meta-crystal (Fig. 3e), reaffirming the effect of meta-grain boundaries in restricting the propagation of shear bands. It is worth noting that although the shear bands in the eight meta-grains appeared to be two diagonal bands, which were similar to those seen in the bi-meta-crystals shown in Fig. 3b, the shear bands were in fact four and impeded by the boundaries before propagating to neighbour meta-grains (Fig. 5b-3), avoiding stress drops. In addition, the distribution of shear bands is substantially more homogeneous throughout the meta-crystals with an increasing number of meta-grains (e.g. Fig. 5e vs a), making the deformation much more stable, explaining why the stress drop is increasingly diminished and ultimately eliminated with increasing the number of meta-grains, resulting in stable flow stress as seen in Fig. 4. Moreover, meta-grain boundaries were also able to temporarily stop the propagation of shear bands, hence strengthening the meta-crystal during plastic deformation. The higher the boundary area the higher the induced hardening. Consequently, the hardening rate increases with reducing the size of meta-grains (Fig. 4b, c).

**Boundary coherency: effect of strut connectivity at boundaries**. For polygrain alloys, it is suggested that the coherency of lattice at grain boundaries is one of the governing factors in the slip transmission, and hence the strengthening induced by grain boundaries[35,36]. Although the nature of bonding in crystals and meta-crystals is different, namely the bonding between atom-like nodes is made by physical struts while the connection in intrinsic crystals is by atomic bonds, the coherency of the lattice at boundaries is still expected to be influential in meta-crystals. In particular, the change in lattice orientation across a meta-grain boundary disrupts the continuity of strut arrangements and might cause open-ended struts if not being deliberately connected by computer design. This might significantly weaken the polygrain-like meta-crystals. To minimise such detrimental effect, 2D frames were introduced at boundaries to ensure the connectivity of struts to meta-grain boundaries (see Fig. 5 and Supplementary Fig. 4). Although the coherency at meta-grain boundaries is better represented by coincident site lattice as defined for intrinsic polycrystals[35,36], the connectivity of struts to an architected boundary was used to reflect the degree of coherency of architected polygrain-like materials. Figure 6a shows

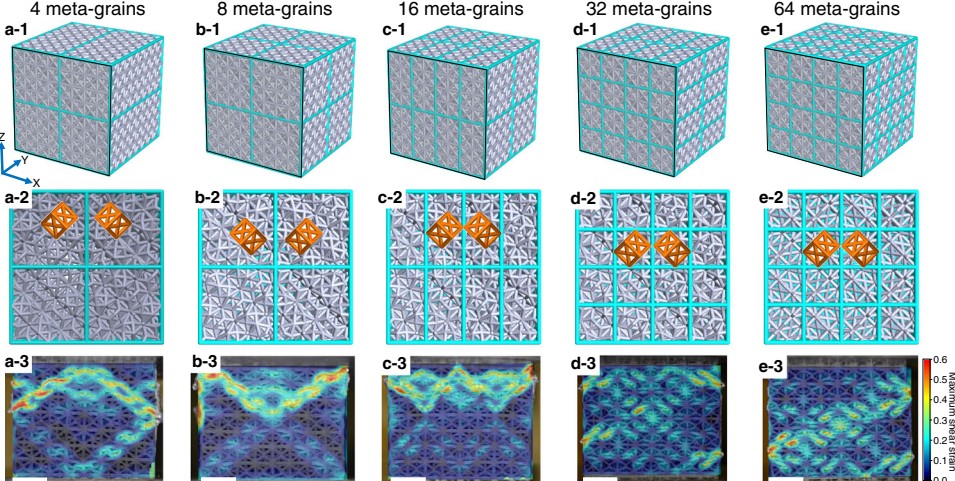

**Fig. 5 Models demonstration and deformation behaviours of polygrain-like meta-crystals.** Strain distribution at 20% strain on meta-crystals with respect to different lattice configurations after digital image correlation analyses, where unit-cell orientation and boundary location are highlighted. **a** 4 meta-grains. **b** 8 meta-grains. **c** 16 meta-grains. **d** 32 meta-grains. **e** 64 meta-grains, where the colour bar indicates the maximum shear strain and the length-scale bar is 10 mm.

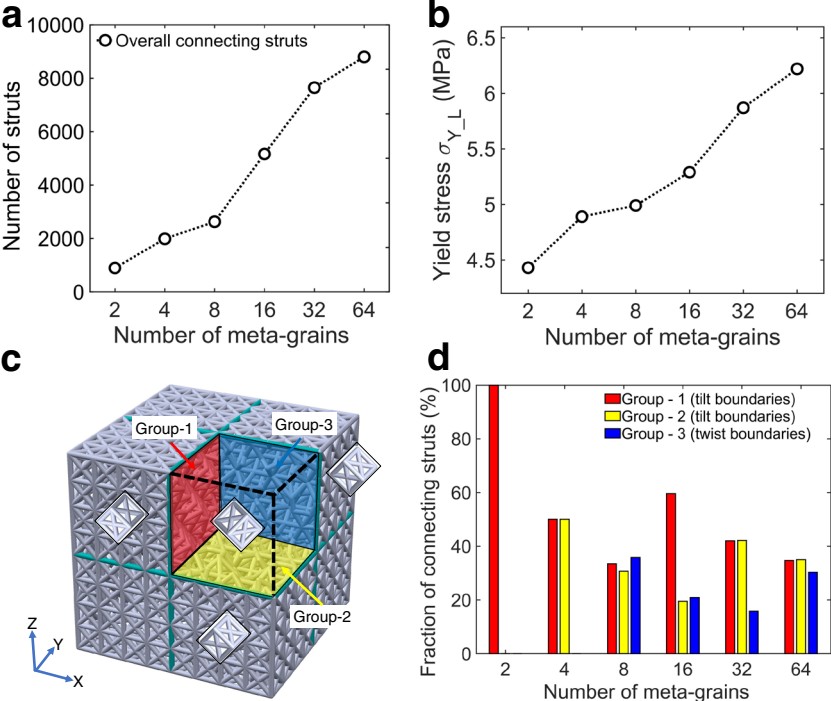

**Fig. 6 Correlating the hardening effects to the boundary coherency and types. a** The total number of struts connecting to the meta-grain boundaries. **b** Measured yield stress for meta-crystals containing different numbers of meta-grains. **c** Schematic diagram illustrating the classification of boundaries with insets showing lattice orientation in each grain. **d** The fraction of struts connected to boundaries of a specific boundary type.

the total number of struts connected to boundaries. It can be seen that the number of struts increased with decreasing the meta-grain size, which was mainly because of the increased boundary areas. With the comparison to the measured yield stress for each meta-crystal shown in Fig. 6b, it is clear that the increasing trend of the yield stress (Fig. 6b) followed a similar trend of the total number of connected struts (Fig. 6a). Therefore, higher connectivity of struts to boundaries made the boundaries better supported, i.e. stronger boundaries, resulting in higher yield strength and effectively impeding shearing propagation.

The boundary connectivity depends on the types of boundary, namely tilt and twist types. According to the lattice orientation relationship used in this study, boundaries in the studied meta-crystals can be classified into three groups: group 1 (parallel to Y–Z plane), group 2 (parallel to X–Y plane) and group 3 (parallel to X–Z plane), where groups 1 and 2 were of the tilt type and group 3 was of the twist type (Fig. 6c). Figure 6d shows the percentage of the number of struts belong to each boundary group in relation to the total struts connecting to boundaries in the whole meta-crystal. Correlating Fig. 6b and d suggests that the introduction of tilt boundaries might induce more hardening effects on yield strength (e.g. 4 meta-grains and 32 meta-grains in Fig. 6b) than that of the twisted boundary (e.g. 8 meta-grains and 64 meta-grains of which the fractions of twist boundary were relatively high). The designed tilt boundary had higher coincident lattice sites (i.e. high coherency) at boundaries than the used twist boundary. Therefore, tilt boundaries were more supported and stronger, explaining higher extents of hardening in the tilt boundaries compared to the twist boundaries. In addition, the alignment of the tilt boundaries in the considered meta-crystals increased the frequency of interaction between the tilt boundaries with shear bands of $\{2\,0\,0\}\langle0\,1\,1\rangle$, making the tilt boundaries more effective in stopping the transmission of shear bands across boundaries, hence more hardening induced by the tilt boundaries.

## Discussion

The activation of a slip system is defined by the crystal lattice, critical resolved shear stress and Schmid factor that relates the slip system to the loading direction[48]. Equivalently, this study shows a shear band in meta-crystals is governed by lattice structures, critical buckling stress and the orientation of lattice with respect to the loading direction. Although there are significant differences between crystals and meta-crystals, the great equivalence enables the translation of some key metallurgical phenomena to the design and control of meta-crystals. The identified shear-band systems, i.e. $\{0\,2\,2\}\langle1\,0\,0\rangle$ and $\{2\,0\,0\}\langle0\,1\,1\rangle$ and in-depth analyses of buckling provide increased confidence in predicting and controlling the shear bands and spatial damage in polygrain-like meta-crystals.

Focus was made to reveal the underlying mechanisms responsible for the strengthening of meta-crystals containing polygrain-like domains (i.e. meta-grains). Specifically, the enhanced stability and strength of meta-crystals during plastic deformation is mainly due to the strength of meta-grain boundary that is associated with the boundary coherency. Similar to the role of coherency at grain boundaries in polycrystalline alloys, higher coherency results in better-supported struts in architected polygrain-like meta-crystals. In particular, a higher number of total struts connecting to boundaries leads to higher strength. In addition to the influential role of boundary coherency, meta-grain boundaries are effective in shortening and deflecting the propagation of shear bands thanks to the lattice misorientation across meta-grain boundaries. It is worth noting that the yield strength was measured at a strain offset of 0.2% at which the first shear band already happens and experiences the stopping effect of meta-grain boundaries on the shear-band propagation, explaining the increase in macroscopically measured yield strength with decreasing the meta-grain size. An architected tilt type of boundaries induced higher hardening than an architected twist one, highlighting that different coherency of

meta-grain boundaries results in different degrees of hardening. This implies that high strength architected materials can be designed without reducing the size of meta-grains. This finding is important because it is not always possible to keep reducing the size of meta-grains due to physical constraints. The engineering of boundary coherency similar to the grain engineering approach in metallurgy can enable the achievement of high strength and damage-tolerant architected materials. In addition, as this study only utilised the twin FCC45 boundary, the use of different misorientations and boundary characteristics with good connectivity to the boundary will enable additional freedom in tailoring the properties of architected materials.

The introduction of crystalline-inspired features to break the long-range periodicity in a singly orientated lattice into meta-grains of different orientations inhibits the fast propagation of the shear band and prevents catastrophic failure. It is worth noting that the improvement in the yield strength and post-yield behaviour induced by the polygrain-inspired lattice materials is in relation to single crystal-like ones with the former outperforms the latter one of the same lattice type, such as FCC in this study. Tremendous potential opportunities to strengthen and tailor the properties of meta-crystals remain to be explored when combining the reduction in meta-grain size, meta-grain morphology and distribution and the boundary type (e.g. low versus high-angle boundaries) with different types of lattice (e.g. Kagome and honeycomb instead of FCC). As the origin of the hardening effect results from the stopping and shortening of shear bands induced by the meta-grain boundaries, the boundary hardening approach should be easily translated to other unit-cell types[38,49–51]. In metallurgy, the isotropic behaviour can be achieved by an aggregate of randomly oriented grains[52,53]. Most of the lattices are inherently anisotropic due to the directionality in bondings. The anisotropy is exaggerated for metallic structures fabricated by additive manufacturing due largely to columnar grain growth[54,55]. Considering the high anisotropy of lattice materials, increasing the number of domains of different crystallographic orientations, i.e. the polygrain-like meta-crystals, not only can increase the strength and energy absorption but also minimise the anisotropy of architected lattice materials. Such minimised anisotropy can improve the material performance in multiaxial and complex loading paths.

## Methods

**Meta-crystal design**. A cubic unit cell with dimension of $5 \times 5 \times 5$ mm$^3$ and strut with a diameter of 1 mm (shown in Supplementary Fig. 1) inspired by FCC atomic lattice was designed using nTopology Element. A global orthogonal coordinate system was right-handed with $Z$ axis being parallel to the build direction of fabrication and also the loading direction of the compression test. To study the orientation dependence, three architected materials containing singly orientated lattices of the FCC unit cell (i.e. equivalent to single crystals) were designed. The first single meta-crystal was generated in two rotation steps as follows: the three $\langle 1\,0\,0 \rangle$ directions of the unit cell were initially aligned parallel to the global $X$-, $Y$- and $Z$ axes. The unit cell was then rotated 45° clockwise about the global $X$ axis and followed by 30° clockwise about the global $Y$ axis. The first single meta-crystal was then constructed by infilling a global cube with a side length of 40 mm using the rotated unit cells. Similarly, the other two single meta-crystals were created following the same procedure for the first rotation, however with a second rotation of 45° and 60° about the global $Y$ axis. In total, three singly orientated meta-crystals with their [1 0 0] inclined at different angles to the loading direction were created, referred to as FCC30, FCC45 and FCC60 depending on the unit-cell rotation about the $Y$ axis.

The same design approach as presented in our previous study was used to create polygrain-like architected materials called meta-crystals[29]. The global lattice structure with dimensions of $40 \times 40 \times 40$ mm$^3$ was divided into domains called meta-grains of which the lattice orientation was tailored to be different to that of its adjacent meta-grains. For example, for a meta-crystal containing two twinned meta-grains (Fig. 3e), the unit cell in the left meta-grain was constructed in the same rotation way for the FCC45 (Figs. 1a and 3c) while the one in the right meta-grain followed the same rotation step, but in the anticlockwise rotation direction. Note that the rotation of the internal structure in each meta-grain leads to incomplete unit cells in the vicinity of the meta-grain boundaries and therefore

generates open struts that would weaken a meta-crystal at meta-grain boundaries, a planar frame of a 2D face-centred square unit cell with a side length of 5 mm was inserted at the boundary between two meta-grains. This allows for the open struts to connect to the boundary and maintain the connectivity. One such frame structure at a boundary is highlighted in Supplementary Fig. 4. Meta-crystals containing 4, 8, 16, 32 and 64 meta-grains were constructed with boundaries as highlighted (Fig. 5). It is noted that, with the consideration of the size of the FCC unit cell (Fig. 1a), dividing meta-crystal into 64 meta-grains is nearly the finest strategy to enable one meta-grain to contain one complete FCC unit cell. The size of a meta-grain was calculated as the diameter of a sphere that has the same volume as the meta-grain.

**Experiments and post analysis**. The designed meta-crystal models were fabricated using fused deposition modelling (FDM) 3D printer (Ultimaker 2) with polylactic acid (PLA) filament obtained from RS components. Some main printing parameters were given as follows: layer thickness of 0.1 mm, printing speed of 30 mm · s$^{-1}$, nozzle travel speed of 120 mm · s$^{-1}$, nozzle diameter of 0.4 mm, printing temperature of 210 °C and build plate temperature of 60 °C. The build direction was parallel to the $Z$ axis shown in Fig. 2.

Compression tests were carried out by a 100 kN Zwick testing machine at a strain rate of 0.001 s$^{-1}$ at room temperature. Lubricant (dry molybdenum disulphide) was used to minimise the effect of friction between lattices and compression plates. The loading direction was parallel to the $Z$ axis shown in Fig. 1a. Nominal engineering stresses were calculated by dividing the recorded forces by the nominal cross-section area ($41 \times 41$ mm$^2$). Engineering strains were derived by dividing the change in the length along the $Z$ direction by the initial length 41 mm. Images of the front face of the meta-crystal (parallel to global $X$–$Z$ plane shown in Fig. 5) were taken at 1 s intervals by a Nikon D7100 camera with 200 mm Nikkor macro lenses. The series of captured images during the deformation were analysed by Digital Image Correlation (DIC) via a commercial software DaVis with the following settings: image size of $6000 \times 4000$ pixels, subset size of $101 \times 101$ pixels and a step size of 25 pixels.

The mechanical properties of the base material were obtained from the uniaxial tensile tests on specimens designed according to ISO 527 standard. Specimens were fabricated by FDM using the same printing parameters as the meta-crystals. Average elastic modulus $E_S$ and yield stress $\sigma_{Y\_S}$ of the base material were calculated from three tensile stress–strain curves, giving $E_S = 1190.4$ MPa and $\sigma_{Y\_S} = 47.6$ MPa.

**Buckling force analyses**. To understand the deformation behaviours of lattices under loading, we performed force analyses on the struts of FCC unit cells orientated to different directions with respect to the loading direction. The directions of individual struts were defined with respect to the local coordinate system of the unit cell (shown in Fig. 2a). To analyse a certain strut, the compressive force $F$ was assumed to be parallel to the $Z$ axis which is subsequently resolved into two orthogonal forces: $F_{ra}$ (along the axis of strut) and $F_{rp}$ (perpendicular to $F_{ra}$ and within the plane formed by the axis of strut and global $Z$ axis). Since the buckling is the main failure mode of slender struts, critical buckling force $F_{cr}$ is calculated through Eq. (2) from Euler's buckling criteria[56], where $E_s$ is the elastic modulus of base material, $I$ is the second moment of area, $K$ is the strut effective length factor that is dependent on the end fix of struts ($K$ should be almost the same for every strut due to the same end fix), and $L$ is the length of the strut. For inclined strut, the critical buckling force along global $Z$ axis was $F = F_{cr}/\sin\alpha$, where $\alpha$ is the angle between the $Z$ axis and the direction of $F_{rp}$. Particularly, the compressive force $F$ for [1 0 0] and [0 1 0] struts can be obtained from Eqs. (3) and (4), where the rotation angle $\theta$ is equivalent to the angle between the global $Z$ direction and the [0 $\frac{\sqrt{2}}{2}$ $\frac{\sqrt{2}}{2}$] strut (shown in Fig. 1). In addition, the buckling tendency of struts can be determined by the slenderness ratio $S = KL/R_g$, where $R_g$ is the radius of gyration of the cross-section of struts. Higher $S$ indicates higher susceptibility to buckling.

$$F_{cr} = \frac{\pi^2 E_s I}{(KL)^2} \tag{2}$$

$$F_{[100]} = \frac{F_{cr}}{\sin(\theta)} \tag{3}$$

$$F_{[010]} = \frac{F_{cr}}{\cos(\theta)\cos(\frac{\pi}{4})} \tag{4}$$

**Finite element analysis (FEA)**. Finite element analysis (FEA) was carried out to study the spatial deformation behaviour of internal architected struts in single meta-crystals using Abaqus/Explicit to assist the DIC analyses. Two FEA lattice models for FCC45 and FCC60 were designed with the dimensions for the unit cell and 2D frame remained the same (as for the meta-crystal design), but a reduced global dimensions of $20 \times 20 \times 30$ mm$^3$ to reduce the computational time (Supplementary Fig. 3). FEA lattice models were meshed using B31 Timoshenko beam elements with an approximate seed size of 2 mm. The elemental beam orientation

was assigned to be along the direction of the strut. Materials properties for lattice in FEA were obtained from the uniaxial tensile tests of 3D-printed specimens. Two rigid plates were placed on the top and bottom of the FEA models and meshed by R3D4 rigid elements with seeds size of 2 mm. Tie constraints were applied for top and bottom lattice nodes that were in contact with the top and bottom plates, respectively. The bottom plate was fixed while the top plate moved down for compression with the strain speed adjusted using a smooth step amplitude.

## Data availability

The datasets related to this study are available from the corresponding author upon reasonable request.

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

## Acknowledgements

The authors thank financial support via the Excellent Funds for Frontier Research provided by Imperial College London.

## Author contributions

C.L.: conceptualisation, methodology, software, formal analysis, investigation, data curation, writing—original draft, writing—review and editing. J.L.: formal analysis, investigation, writing—review and editing. M.S.P: conceptualisation, resources, formal analysis, supervision, funding acquisition, writing—review and editing.

## Competing interests

The authors declare no competing interests.
