## [Peer Review File · Nature Communications]

REVIEWER COMMENTS

Reviewer #1 (Remarks to the Author):

In this paper, the authors designed the poly-grain truss-based lattices with different grain boundaries and fabricated the designed lattices via 3D printing. The authors further investigated the shear banding behaviors of poly-grain lattices and relevant buckling of struts via DIC and FEM, and found that the grain boundary can block the propagation of shear bands, leading to the significant strengthening of poly-grain lattices. These results are novel and interesting, and would have significant impact on the design and fabrication of architected lattices with high strength. The manuscript is well-written and well-organized. Therefore, the reviewer can recommend this paper for publication. But the authors have to address the following points to improve the current manuscript.

1. For the lattices, the relative density is an important parameter. The stiffness and strength of lattice are significantly dependent on its relative density. Although the stress data is normalized by the relative density in the current manuscript, it is still necessary to provide the relative density of all fabricated poly-grain lattices. It is helpful for the reader to know the change of relative density with the variation of grain boundary.
2. The authors observed the shear banding of poly-grain lattices with the unit cell size of millimeter. Such shear banding is related to the buckling of single struts in the lattice. Previous studies (Nano Lett, 18, 4247-4256, 2018) about high-entropy alloy-polymer nanolattices exhibited that the buckling of struts propagates from the bottom to top during compression, which is different from the shear banding in the current manuscript. The authors should compare these two deformation modes between millimeter-size lattices and nanolattices.
3. Recent perspective (Nature Mater, 15, 373, 2016) showed that for mechanical metamaterials, the smaller is the stronger. It is suggested for the authors to add a discussion about the influence of unit cell size and strut size on the strength and the shear banding behaviors.
4. In the lines 230-231 of page 10, the authors stated that "...the hardening rates became more stable during deformation with a reduction in meta-grain size." This phenomenon is very interesting. The authors are suggested to add a physical or mechanical explanation on this phenomenon.
5. Recent studies (PNAS, 116, 6665-6672, 2019) reported that the mechanical properties of iso-truss lattice are better than those of octet-truss lattice. The authors are suggested to introduce the iso-truss unit cell in the introduction section, when the authors mentioned the unit cell types in the lines 53-54 of page 3. Recent review literatures (Adv Mater, 29, 1701850, 2017; MRS Bulletin, 44, 758-765, 2019; Small, 16, 1902842, 2020) summarized the recent advance of lattice materials. The authors should introduce these relevant review papers and summarize the recent development of lattice materials.

Reviewer #2 (Remarks to the Author):

This study takes inspirations from metallurgical grain boundary hardening mechanisms in crystalline materials to guide the design of shear-band formation in the deformations of architected

metamaterials. They found that tilt type of boundaries induced higher hardening than an architected twist one, highlighting that different coherency of meta-grain boundaries results in different degrees of hardening. The obtained insights can facilitate the development of higher strength architected materials with controlled damage path. Both experimental and simulations were conducted to correlated with their hypothesis.

This study is a follow-on from the authors' previous paper, Pham, M. S., Liu, C., Todd, I. & Lertthanasarn, J. Damage-tolerant architected materials inspired by crystal microstructure. *Nature* 565, 305–311 (2019), which reports similar hypothesis.

While the analogy used by the authors can certainly facilitate understanding of shear band formation in architected lattice materials, the reviewer is struggling with the contribution of the current study to the field of architected material design in general.

1. Specifically, are designs reported in the current study applicable to other types of architectures? e.g., bend dominated topology or architectures with different nodal connectivity (octet-truss lattice, Kagome lattices)?
2. How about the density of strength and stiffness scaling of these architectures compared to other reported topologies?
3. How are the reported finding compare with other topologies in energy absorptions and post-yielding behaviors (e.g., honeycomb or close-cell foams)?
4. The current study may only be applicable to loading in a particular direction. It may not be valid when loading is applied in other directions (slightly off axis or triaxial loading)
5. The stress-strain curves reported in Fig. 1 and Fig. 3 don't seem to have significant differences in strength and curve shape differences --- the manufacturing defects in FDM could easily dominate the variations of stress-strain curves as compared to the meta-crystal effects. Additionally, printing directions and anisotropic effects in FDM can also be a more significant factor than the meta-crystal effects. As a result, it is not very convincing how applicable the finding in practical 3D printed lattice materials.

Reviewer #3 (Remarks to the Author):

This study reports the shear banding behavior in FCC meta-crystal lattices. This is a continued effort based on the authors' previous work (*Nature* 565 (2019): 305). The authors observed a single shear band in the meta-single crystal lattice but different slip systems were found in differently oriented meta-

crystal. In contrast, a boundary strengthening behavior and even multiple shear bands were revealed in the meta-polycrystal, suggesting the role of the meta-grain boundary in strengthening of meta-crystal lattices. The authors also studied the critical effect of boundary coherency on the strengthening of these meta-crystal lattices. In general, I think the finding of the boundary coherency effect is very interesting, despite not surprising in understanding the origin of the strengthening in the meta-crystals. I have the following questions/comments:

Major questions:

1. In the bi-meta-crystal, what are the orientations of the two meta-grains? The authors studied the highly localized single shear banding behavior in the FCC30, FCC45, and FCC60 first in single meta-crystals before studying bi-meta-crystal. Did they use two different orientations of FCC30, FCC45, FCC60, or two twinned & coherent FCC45? The Methods section shows twinned FCC45? It is not quite clear and please clarify. In order to show the local misorientation effect, I would think studying two different orientations would be more representative for a general incoherent grain boundary trend.
2. The authors experimentally disclosed the interactions between shear band(s) and the meta-grain boundaries in the meta-crystals. While the DIC-based direct observation of the shear banding activity is interesting, the main results in Fig.3 & Fig. 4 were already reported in their prior work (main text and Extended Data, Nature 565 (2019): 305). The diffused shear banding behavior and the underlying mechanism in the bi-meta-crystal were also discussed before and the results in "Line 168-238" are representation of their prior work. From architecture optimization viewpoint, it would be of interest to offer some insights or demonstrate how the diffused shear banding, essentially distinct buckling paths in different meta-grain orientations, could be "engineered" in a meta-polycrystal by architecture design or topology optimization.
3. In crystalline metals, depending on the local misorientation between adjacent grains, low-angle grain boundary or high-angle grain boundary could arise. The misorientation angle between the two meta-grains can play a critical role in governing the mechanical behavior. Key information seems missing here towards a quantitative understanding of meta-grain boundary misorientation angle on the shear band propagation. For example, what would the bi-meta-crystals of FCC30+FCC45, FCC30+FCC60, FCC45+FCC60 behave, respectively? Unlike the dislocation slip in the FCC-crystalline solids, the shear band system is highly load-orientation dependent in the FCC meta-crystal, for example, (022) \langle 100 \rangle in FCC30 and (200) \langle 011 \rangle in FCC45 and FCC60. This is closely related to the buckling of the constituent struts under a given crystal orientation (Eqs 1-3). One may think of numerous buckling scenarios of FCC35, FCC36, FCC37, etc in a meta-polycrystal. As such, the configurations of the shear band activity in a meta-polycrystal consisting of many differently-oriented meta-grains would be complex to predict. This shear banding behavior is fundamentally different from the classical well-defined slip system in FCC crystalline solids, largely due to the nature of atomic motion during slip in a crystalline solid. I am concerned how well the crystallographic metallurgy lessons can be applied here to design these meta-crystal materials.
4. In the last section of Results, the authors studied the boundary coherency effect via adjusting struct connectivity in order to mimic the interfacial coherency in crystalline solids. Specifically, the twist boundary strengthening was found not as robust as the tilt boundary strengthening, in which the higher coincident lattice sites essentially lead to higher coherency. In the design, they added 2D frames at the boundary to ensure the connectivity of struts and to avoid open-ended struts within meta-grain

boundaries. Please comment on the morphology difference of the 2D frames of different boundary groups with different coherency levels.

5. The last critical question I have is the “normalized stress” (stress normalized by relative density) plot for various Meta-grains. The authors found a least square linear fitting gives Hall-Petch relationship to describe the boundary strengthening in meta-crystals. They argued that normalizing the strength by relative density for the lattices with different meta-grain numbers can eliminate the effects of relative density since the different numbered meta-grains have different relative densities. Nevertheless, as well-known, the specific strength of architected materials generally do not scale linearly with relative density, but often comes with an exponential relationship with a scaling factor >1 (except for the Octet lattice, e.g., PNAS, 112 (37), 11502, 2015). Normalizing the relative density cannot fully eliminate the coupled effect of increased density induced strengthening.

Minor questions:

1. Line 113, what are FCC 30, FCC45, and FCC60? I see the description in the Methods section. Please add a reference to “Methods” here.
2. Line 217, in the conventional H-P relationship, σ_0 is the frictional stress, what is the physical meaning of σ_0 in meta-crystal? Have the authors compared σ_0 with the yield strength of the meta single-crystal?

Referees' comments:

Referee #1 (Remarks to the Author):

In this paper, the authors designed the poly-grain truss-based lattices with different grain boundaries and fabricated the designed lattices via 3D printing. The authors further investigated the shear banding behaviours of poly-grain lattices and relevant buckling of struts via DIC and FEM, and found that the grain boundary can block the propagation of shear bands, leading to the significant strengthening of poly-grain lattices. These results are novel and interesting, and would have significant impact on the design and fabrication of architected lattices with high strength. The manuscript is well-written and well-organised. Therefore, the reviewer can recommend this paper for publication. But the authors have to address the following points to improve the current manuscript.

1. For the lattices, the relative density is an important parameter. The stiffness and strength of lattice are significantly dependent on its relative density. Although the stress data is normalised by the relative density in the current manuscript, it is still necessary to provide the relative density of all fabricated poly-grain lattices. It is helpful for the reader to know the change of relative density with the variation of grain boundary.

Response 2:

The relative densities both calculated from CAD models and experimentally measured from fabricated specimens were provided in Table S1. The relative density slightly increases with the numbers of meta-grains, but the variation was relatively small.

2. The authors observed the shear banding of poly-grain lattices with the unit cell size of millimetre. Such shear banding is related to the buckling of single struts in the lattice. Previous studies (Nano Lett, 18, 4247-4256, 2018) about high-entropy alloy-polymer nanolattices exhibited that the buckling of struts propagates from the bottom to top during compression, which is different from the shear banding in the current manuscript. The authors should compare these two deformation modes between millimetre-size lattices and nanolattices.

Response 3:

Revision was made to discuss the different deformation behaviours (Page 8, first paragraph) with proper references.

3. Recent perspective (Nature Mater, 15, 373, 2016) showed that for mechanical metamaterials, the smaller is the stronger. It is suggested for the authors to add a discussion about the influence of unit cell size and strut size on the strength and the shear banding behaviours.

Response 4:

Discussions were added about the effect of length scale of struts in reference to the mentioned perspective paper (Page 8, first paragraph), the size of the unit cell and struts (Page 8, first paragraph) and relative densities (Page 11, last paragraph) with proper references. In addition, since we studied the deformation of meso-structure, discussions on the shear behaviours of other meso-scaled architected lattice materials reported in literature were also provided (Page 5, first paragraph).

4. In the lines 230-231 of page 10, the authors stated that "...the hardening rates became more stable during deformation with a reduction in meta-grain size." This phenomenon is very interesting. The authors are suggested to add a physical or mechanical explanation on this phenomenon.

Response 5:

The explanation was provided on Page 9, Page 10 and Page 12. Text was added to expand the explanation (Page 10, last 5 lines; Page 12, second paragraph). The more stable hardening rate is related to the supporting boundaries, misorientation across a boundary and the stopping effect induced by boundaries on the shear band propagation that minimises dominantly localised and fast propagating shear bands.

5. Recent studies (PNAS, 116, 6665-6672, 2019) reported that the mechanical properties of iso-truss lattice are better than those of octet-truss lattice. The authors are suggested to introduce the iso-truss unit cell in the introduction section, when the authors mentioned the unit cell types in the lines 53-54 of page 3. Recent review literatures (Adv Mater, 29, 1701850, 2017; MRS Bulletin, 44, 758-765, 2019; Small, 16, 1902842, 2020) summarized the recent advance of lattice materials. The authors should introduce these relevant review papers and summarize the recent development of lattice materials.

Response 6:

The authors thank the suggestions for the studies and reviews on recent developments in lattice materials. These previous works were introduced and referenced in the Introduction section (Page 3, first paragraph).

Reviewer #2 (Remarks to the Author):

This study takes inspirations from metallurgical grain boundary hardening mechanisms in crystalline materials to guide the design of shear-band formation in the deformations of architected metamaterials. They found that tilt type of boundaries induced higher hardening than an architected twist one, highlighting that different coherency of meta-grain boundaries results in different degrees of hardening. The obtained insights can facilitate the development of higher strength architected materials with controlled damage path. Both experimental and simulations were conducted to correlated with their hypothesis.

This study is a follow-on from the authors' previous paper, Pham, M. S., Liu, C., Todd, I. & Lertthanasarn, J. Damage-tolerant architected materials inspired by crystal microstructure. Nature 565, 305–311 (2019), which reports similar hypothesis.

While the analogy used by the authors can certainly facilitate understanding of shear band formation in architected lattice materials, the reviewer is struggling with the contribution of the current study to the field of architected material design in general.

1. Specifically, are designs reported in the current study applicable to other types of architectures? E.g., bending dominated topology or architectures with different nodal connectivity (octet-truss lattice, Kagome lattices)?

Response 7:

Major revisions were made regarding to the significance of this present study in relation to our own study published in Nature (ref. 28). We particularly revised the Introduction and discussion sections to clearly highlight the significantly distinctive contribution of this study in relation to our previous one. The previous paper presented the overall of the crystal-inspired approach utilising the metallurgical hardening to strengthen architected materials. It reported the hardening phenomena, but it did not provide in-depth discussion on the underlying mechanisms responsible for the strengthening effects, in particular concerning how the meta-grain boundaries can strengthen the meta-crystals, and how boundary characteristics (in

particular coherency) contribute to the strengthening effects. To unravel such underlying mechanisms, this current paper examined the shear behaviour in singly orientated lattice materials, then studied the shear bands in multi-orientated lattice, in particular their interactions with boundaries of meta-grains to shed light onto the strengthening mechanism induced by meta-grain boundaries. Consequently, this current study provides the materials designers new fundamental insights into the boundary strengthening, enhanced confidence and a sound basis towards full utilisation of the meta-crystal approach (in particular the boundary strengthening) to create lightweight and high strength architected materials. Consequently, we believe this study provides new scientific findings with significant technological implications for the design of lightweight, high strength architected materials.

The presented polycrystal-inspired approach proposes to design multiple domains (i.e. meta-grains) of different orientations of a given lattice type instead of creating a singly uniform orientation of the same lattice type. The boundaries between meta-grains play as obstacles to the propagation of localisation/damage, thus generating the strengthening effect. Hence the approach is applicable to any other types of lattice (including Kagome and Octet). We correspondingly made revision the Introduction (Page 3, second paragraph) and Discussion (Page 16, second paragraph) to discuss the applicability of the crystal-inspired approach to other types of lattice.

2. How about the density of strength and stiffness scaling of these architectures compared to other reported topologies?

Response 8:

The relationship between relative densities and strength were provided in Fig. 4d. Discussion on the effect of relative density was also provided (Page 11, second paragraph). Please also refer to our response 16 below.

3. How are the reported finding compare with other topologies in energy absorptions and post-yielding behaviors (e.g., honeycomb or close-cell foams)?

Response 9:

This comment is relevant to the Reviewer's comment 2. Although we did not compare the FCC meta-crystals with other topologies, the enhancement in energy absorption and post-yielding behaviour is seen to increase with the number of meta-grains. It is expected that if the approach was applied to other lattice types (including honeycomb or close-cell foams), a similar enhancement would be observed, e.g. improved performance in a polycrystal-like honeycomb in comparison to the single oriented honeycomb. Please refer to our response 8.

4. The current study may only be applicable to loading in a particular direction. It may not be valid when loading is applied in other directions (slightly off axis or triaxial loading).

Response 10:

We thank the Reviewer for highlighting this important point. We agree that the loading condition can significantly influence the mechanical behaviour of architected lattice materials that are inherently anisotropy due to the directionality in the bonding. This is a reason why we studied the anisotropy of shear band activities in different loading directions with respect to the lattice orientations (Figs. 1 and 2). The anisotropy was shown in the Results section: shear behaviour in singly-orientated meta-crystals upon loading in different directions. The anisotropy is of great important in multi-axial loading. It is well known in the metallurgy that the isotropic behaviour can be achieved by an aggregate of randomly oriented grains. As our approach proposes to strengthen the architected materials by increasing the number of "grains"

of different orientation, the polygrain-like mesostructures should help to increase the isotropic thanks to increased random in lattice orientations, hence improving the behaviour in multi-axial loading. We added a discussion in this revision (Page 16, second paragraph) to indicate that the polygrain-inspired approach can help to minimise the anisotropy found in singly orientated lattice materials thanks to the increase in lattice orientations in meta-crystals. It is worth noting that the extent of effectiveness in reducing the anisotropy and increasing the strength varies with different boundaries types, lattice orientations, the size and morphologies of meta-grains. Therefore, more comprehensive investigations are needed, and these are main topics of our current works.

5. The stress-strain curves reported in Fig. 1 and Fig. 3 don't seem to have significant differences in strength and curve shape differences --- the manufacturing defects in FDM could easily dominate the variations of stress-strain curves as compared to the meta-crystal effects. Additionally, printing directions and anisotropic effects in FDM can also be a more significant factor than the meta-crystal effects. As a result, it is not convincing how applicable the finding in practical 3D printed lattice materials.

Response 11:

We agree with the Reviewer that the processing defects from FDM can result in the variation of the measured strength. In this study we consistently kept the same orientation between the loading and build directions. We conducted 3 repeated tests for lattice models of FCC30, FCC45, FCC60, meta-crystals containing 2, 8, and 32 meta-grains (Supplementary Fig. 5) to ensure that the measured improvement in the strength were due to the decrease in the size of meta-grains. All repeated tests show highly reproducible stress-strain responses that were almost identical, in particular the yielding behaviour. Main variation was seen on the onset of the densification which was not the interest of the current study. This highly reproducible plastic deformation means that the effect associated with the processing quality (including build orientation) was negligible and we can be confident in the contribution induced by the polycrystal-like mesostructure. We added text (Page 10, first paragraph) and provided Supplementary Fig. 5 accordingly.

Reviewer #3 (Remarks to the Author):

This study reports the shear banding behaviour in FCC meta-crystal lattices. This is a continued effort based on the authors' previous work (Nature 565 (2019): 305). The authors observed a single shear band in the meta-single crystal lattice, but different slip systems were found in differently oriented meta-crystal. In contrast, a boundary strengthening behaviour and even multiple shear bands were revealed in the meta-polycrystal, suggesting the role of the meta-grain boundary in strengthening of meta-crystal lattices. The authors also studied the critical effect of boundary coherency on the strengthening of these meta-crystal lattices. In general, I think the finding of the boundary coherency effect is very interesting, despite not surprising in understanding the origins of the strengthening in the meta-crystals. I have the following questions/comments:

Major questions:

1. In the bi-meta-crystal, what are the orientations of the two meta-grains? The authors studied the highly localised single shear banding behaviour in the FCC30, FCC45, and FCC60 first in single meta-crystals before studying bi-meta-crystal. Did they use two different orientations of FCC30, FCC45, FCC60, or two twinned & coherent FCC45? The Methods section shows twinned FCC45? It is not quite clear and please clarify. In order to show the local misorientation

effect, I would think studying two different orientations would be more representative for a general incoherent grain boundary trend.

Response 12:

We thank the Reviewer for the comments. We revised the related text to clarify the model design of the bi-meta-crystal and confirmed that the two twinned FCC45 meta-grains were used for the meta-crystal (Page 8, second paragraph, Methods and Supplementary Fig. 4). We agreed that the study of boundary characteristics, in particular high-angle incoherent boundaries, would provide interesting results – this is our current focus. In this study, we tried to get clean data of the meta-grain size effect, then studied some key aspects of boundary characteristics (coherency induced by tilted and twisted boundary of the same misorientation) before exploring the exciting effects induced by the vast characteristics of boundaries (including variation in misorientations). Such clean data assist us in understanding the contribution from individual factors such as the meta-grain size and coherency.

2. The authors experimentally disclosed the interactions between shear band(s) and the meta-grain boundaries in the meta-crystals. While the DIC -based direct observation of the shear banding activity is interesting, the main results in Fig.3 & Fig. 4 were already reported in their prior work (main text and Extended Data, Nature 565 (2019): 305). The diffused shear banding behaviour and the underlying mechanism in the bi-meta-crystal were also discussed before and the results in "Line 168-238" are re-representation of their prior work. From architecture optimization viewpoint, it would be of interest to offer some insights or demonstrate how the diffused shear banding, essentially distinct buckling paths in different meta-grain orientations, could be "engineered" in a meta-polycrystal by architecture design or topology optimization.

Response 13:

We revised the text in the Introduction section to clearly differentiate this paper from previous paper (Page 3, second paragraph and Page 4, first paragraph). The current paper went much deeper to understand the origin and mechanisms responsible for the boundary strengthening induced by the polygrain-like mesostructure. While we re-used some minimal results to highlight the improved strength and stability (Fig. 3c, e, and some curves in Fig. 4a and black data points in Fig. 4b of this study) from our previous study, significantly new and more in-depth results were presented to help revealing the formation and activities of shear bands (Fig. 1 and 2) and discussing mechanisms that underlie the hardening effect not only seen on the plastic yield, but also on the hardening rate during plastic deformation due to interaction between the boundaries and shear bands (Fig. 3b, d and f, Fig. 4b, c and d, Fig. 5 and corresponding text – most notably on Page 10 and 12). A whole new section focused on the effect of the boundary coherency of tilt and twist boundary (Fig. 6 and corresponding text in the "Boundary coherency" section). Such detailed examinations and analyses are crucial for topology engineering and optimisation guided by metallurgical principles. Please also refer to our response presented in the response 7.

3. In crystalline metals, depending on the local misorientation between adjacent grains, low-angle grain boundary or high-angle grain boundary could arise. The misorientation angle between the two meta-grains can play a critical role in governing the mechanical behavior. Key information seems missing here towards a quantitative understanding of meta-grain boundary misorientation angle on the shear band propagation. For example, what would the bi-meta-crystals of FCC30+FCC45, FCC30+FCC60, FCC45+FCC60 behave, respectively? Unlike the dislocation slip in the FCC-crystalline solids, the shear band system is highly load-orientation dependent in the FCC meta-crystal, for example, (022)⟨100⟩ in FCC30 and

(200)(011) in FCC45 and FCC60. This is closely related to the buckling of the constituent struts under a given crystal orientation (Eqs 1-3). One may think of numerous buckling scenarios of FCC35, FCC36, FCC37, etc in a meta-polycrystal. As such, the configurations of the shear band activity in a meta-polycrystal consisting of many differently-oriented meta-grains would be complex to predict. This shear banding behavior is fundamentally different from the classical well-defined slip system in FCC crystalline solids, largely due to the nature of atomic motion during slip in a crystalline solid. I am concerned how well the crystallographic metallurgy lessons can be applied here to design these meta-crystal materials.

Response 14:

We agreed that the study of more different combinations such as FCC30+FCC45, FCC30+FCC60, FCC45+FCC60) will provide additional understandings of the behaviour of meta-crystals. As we responded in the response 12, this study tries to obtain a clean data of the size effect of meta-grains first before exploring the synergistic effects of size and boundary characteristics (in particular different mis-orientations). As highlighted by the reviewer that the buckling-induced shear band initiation is highly orientation dependence, we studied the shear band activity in different loading directions. In the text (first paragraph, page 5), we stated that there were at least two shear band systems. In response to the Reviewer's comment on a possibility of different shear band systems when varying the lattice orientation from FCC30 to FCC45. We carried out additional work that studied the tendency to buckling (with focus on the $\langle 001 \rangle$ struts) when rotating the orientation of the FCC unit cell from 20° to 70° (Fig. 2b). The additional work found that the $\{0\ 2\ 2\}\{1\ 0\ 0\}$ maintained up to 35.26° , then $\{2\ 0\ 0\}\{0\ 1\ 1\}$ was active. At 35.26° , the two systems were seen to be both active. We printed and tested the FCC35.26 to verify this finding. Supplementary Fig. 2 shows that the two systems were activated for the FCC35.26, confirming the validity of the use of buckling criteria to predict the shear band activity in FCC meta-crystals. First paragraph in Page 6 was revised accordingly.

We acknowledge that the shear band activities in polygrain-like meta-crystals are complex. This is the reason why we carried out this study to start obtaining some in-depth insights that help to build a more complete understanding of shear bands. Although there are distinctive differences in intrinsic crystals and meta-crystals, our studies (including the previous one) demonstrate great similarities in particular regarding the shear band activities. The activation of a slip system is defined by the crystal lattice, critical resolved shear stress and Schmid factor that relates the slip system to the loading direction (ref. 46). Equivalently, our studies show a shear band in meta-crystals is governed by the lattice structure, a critical stress (such as buckling stress in the current study) and the orientation of lattice with respect to the loading direction. Such similarities enable the translation of key metallurgical phenomena to the design of high performance architected materials. In fact, our previous study and this study provide evidence showing that we can translate some key hardening mechanisms in metallurgy (such as the boundary hardening) in crystals to improving the strength of meta-crystals. In addition to the new results (Fig. 2b and Supplementary Fig. 2), we added some texts to reflect the response in the discussion (Discussion, Page 15).

4. In the last section of Results, the authors studied the boundary coherency effect via adjusting struct connectivity in order to mimic the interfacial coherency in crystalline solids. Specifically, the twist boundary strengthening was found not as robust as the tilt boundary strengthening, in which the higher coincident lattice sites essentially lead to higher coherency. In the design, they added 2D frames at the boundary to ensure the connectivity of struts and

to avoid open-ended struts within meta-grain boundaries. Please comment on the morphology difference of the 2D frames of different boundary groups with different coherency levels.

Response 15:

The related text was revised to clearly illustrate the introduced 2D frames (Page 13 – first paragraph, Methods – Meta-crystal design, and Supplementary Fig. 4). The 2D frames of a face centred square with the side length of 5 nm were consistently used for the two types of boundaries.

5. The last critical question I have is the “normalized stress” (stress normalized by relative density) plot for various Meta-grains. The authors found a least square linear fitting gives Hall-Petch relationship to describe the boundary strengthening in meta-crystals. They argued that normalizing the strength by relative density for the lattices with different meta-grain numbers can eliminate the effects of relative density since the different numbered meta-grains have different relative densities. Nevertheless, as well-known, the specific strength of architected materials generally do not scale linearly with relative density, but often comes with an exponential relationship with a scaling factor >1 (except for the Octet lattice, e.g., PNAS, 112 (37), 11502, 2015). Normalizing the relative density cannot fully eliminate the coupled effect of increased density induced strengthening.

Response 16:

We thank the Reviewer in pointing this out and agree that the normalisation of stress by the relative density could not eliminate all the effects induced by the relative density. Nevertheless, the normalisation is often used in literature in attempting to minimise the effects. The text was revised accordingly (Page 10, first paragraph). In addition, Fig. 4d shows that the scaling relationship of relative strength with respect to the relative density quite follows a linear relationship with a pre-factor close to a theoretically predicted lower bound of 0.3, indicating a modest dependence on the relative density. Because all the lattice parameters were unchanged and only lattice orientation was varied, the relative density of meta-crystals is expected to only slightly change with varying the meta-grain size. In fact, Table S1 shows the relative density only slightly increased with increasing the number of meta-grains. Therefore, the normalisation of strength by relative density could significantly minimise the contribution from relative density and better highlight the contribution from boundary hardening. We revised the manuscript (Page 11, last paragraph, and Fig. 4d) to reflect this point.

Minor questions:

1. Line 113, what are FCC30, FCC45, and FCC60? I see the description in the Methods section. Please add a reference to “Methods” here.

Response 17:

Text was revised now (Page 4, third paragraph) and properly referred to the corresponding description in the Methods and the Supplementary Fig. 1.

2. Line 217, in the conventional H-P relationship, σ_0 is the frictional stress, what is the physical meaning of σ_0 in meta-crystal? Have the authors compared σ_0 with the yield strength of the meta single-crystal?

Response 18:

Revision was made to indicate the physical meaning of σ_0 (Page 10, first paragraph). The σ_0 represents the macroscopic critical stress to initiate the plastic yield of a single FCC45 meta-crystal with nominally infinite size.

REVIEWERS' COMMENTS

Reviewer #1 (Remarks to the Author):

I am delighted to read the revised manuscript and the rebuttal letter. I felt that the authors have addressed nearly all the concerns and comments from all three reviewers, and that the manuscript has been indeed improved much. The current study provides a fundamental understanding of strengthening mechanisms of boundary in architected materials, which would have impact on the development of architected materials with high strength. Therefore, I recommended this paper for publication now.

**

I have read the responses to reviewer #2 and the correspond revision parts in the revised manuscript, and felt that the authors have addressed nearly all concerns from the reviewer #2. Especially, the authors have highlighted the significance of the current study, which is distinct from their Nature paper. Although the current manuscript still has few unclear point about the influence of printing parameters and printed defects on the mechanical behaviors, the current study are novel in the field of mechanical metamaterials. Therefore, the current manuscript can be recommended for publication.

Reviewer #3 (Remarks to the Author):

The authors have well addressed all the reviewers' concerns. Their responses are convincing. That being said, I have a few additional suggestions the authors should consider to further improve their paper. I recommend publication after the authors consider these minor revisions.

1.The authors show that localized shear band can develop and propagate easily in a singly oriented lattice, leading to the significant stress drop and softening behavior in the plastic regime. In contrast shear band propagation is inhibited in a poly meta-crystal lattices, giving rise to the hardening behavior. In fact, this is reminiscent of a similar phenomenon in some prior comparative studies on architected materials with periodic unit cells versus stochastic unit cells. It is suggested that the authors include some discussions on the relevant behavior, for example, Acta Materialia, 73 (2014): 259-274.

2.In the introduction, “an periodic arrangement” => “a periodic arrangement”

**

I have completed the review of the authors' responses to Reviewer 2 and I personally think the revised manuscript has addressed the reviewers' comments/concerns satisfactorily.